# *HyPRP1*, A Tomato Multipotent Regulator, Negatively Regulates Tomato Resistance to Sulfur Dioxide Toxicity and Can Also Reduce Abiotic Stress Tolerance of *Escherichia coli* and Tobacco

Xueting Chen [1,2,3,†], Lulu Wang [1,2,3,†], Yan Liang [1,2,3], Xiaomeng Hu [1,2,3], Qianqian Pan [1,2,3], Yin Ding [1,2,3] and Jinhua Li [1,2,3,*]

1   Affiliation State Cultivation Base of Crop Stress Biology for Southern Mountainous Land of Southwest University, Beibei, Chongqing 400715, China
2   Academy of Agricultural Sciences, Southwest University, Beibei, Chongqing 400715, China
3   Key Laboratory of Horticulture Science for Southern Mountainous Regions, Ministry of Education, College of Horticulture and Landscape Architecture, Southwest University, No. 2 Tiansheng Road, Beibei, Chongqing 400715, China
*   Correspondence: ljh502@swu.edu.cn; Tel.: +86-23-68250974; Fax: +86-23-68251274
†   These authors contributed equally to the work.

**Abstract:** Abiotic stresses have led to an extensive decline in global crop production and quality. As one of the abiotic stress factors, sulfur dioxide ($SO_2$) causes severe oxidative damage to plant tissues. Based on our previous study, a tomato hybrid-proline-rich protein 1 (HyPRP1) was found to be involved in abiotic stress and $SO_2$ metabolism, though the gene functions remained largely unknown. In this study, the function analysis of the *HyPRP1* gene was extended, and DNA methylation analysis, subcellular localization, and *cis*-element analysis were performed to investigate the features of this gene. The DNA methylation analysis implied that the *HyPRP1* gene was hypermethylated and the methylation density in the leaf differed from that in the flower and fruit. Subcellular localization analysis identified HyPRP1 localized in the cytoplasm and plasma membrane in vivo. The *E. coli* cells harboring *SlHyPRP1* showed reduced salt and drought resistance. In tomato, when $SO_2$ toxicity occurred, the *HyPRP1* RNAi knockdown lines accumulated more sulfates and less hydrogen peroxide ($H_2O_2$) and showed minimal leaf necrosis and chlorophyll bleaching. In tobacco, the overexpression of *HyPRP1* reduced tolerance against salt stresses exerted by NaCl. We conclude that the heterologous expression of tomato *HyPRP1* in *E. coli* and tobacco reduces abiotic stress tolerance and negatively regulates the resistance to sulfur dioxide toxicity by scavenging $H_2O_2$ and sulfite in tomato.

**Keywords:** sulfur dioxide toxicity; abiotic stress; hybrid proline-rich protein; tomato

## 1. Introduction

Abiotic stresses imposed on plants affect growth and yield. Plants are sessile organisms and are naturally exposed to multiple abiotic stresses during their lifespan, thereby greatly reducing productivity and threatening global food security. Human anthropogenic activities and global climatic changes have exacerbated the negative effects of abiotic stresses on crop productivity [1]. Plants exhibit an adaptive defense response against abiotic stresses through the process of long-term evolution [2]; nevertheless, about 50% of the world crop yield is impacted by abiotic stresses, including drought, high salinity, flooding, extreme temperature (heating, cold or freezing), and ambient air pollution [3].

Sulfur dioxide ($SO_2$) is an air pollutant that has deleterious effects on animal and plant health. In plants, low doses of $SO_2$ toxicity lead to visible effects, such as chlorosis and necrosis, resulting in long-term yield reduction [4,5]. The $SO_2$ directly causes oxidative stress by enhancing production of reactive oxygen species (ROS) [6]. The $SO_2$ readily reacts

with water to form sulfite strong nucleophiles, which can cause harmful reactions with various cellular components, direct damage to plants, and affect human health [5]. Sulfite can be converted by oxidation to sulfate and by deoxidation to produce hydrogen sulfide. The oxidation process is catalyzed by the molybdenum cofactor-containing enzyme sulfite oxidase (SO; EC 1.8.3.1) and the deoxidation process is catalyzed by sulfite reductase (SiR; EC 1.8.7.1) through a process that transfers six electrons of ferredoxins (Fds) [7]. Plants have evolved to adapt to adverse environments; such adaptations include changes in morphology, photosynthesis, antioxidant enzyme activities, and abiotic resistance related genes to cope with unfavorable conditions. In general, plant exposure to $SO_2$ has a negative effect on these processes. Plant physiological activities decline in several days after exposure to $SO_2$. Morphological and biochemical activities are also negatively affected by extensive $SO_2$ exposure [8].

Sustainable and equitable global food security depends at least in part on the development of crop plants with increased resistance to abiotic stresses. Using genetic engineering to improve plant abiotic stress resistance is better than conventional breeding because of its ability to modify a target gene of interest within the same or any other species [2]. Plant hybrid proline-rich proteins (HyPRPs) are putative cell wall proteins enriched with proline and composed of a repetitive proline-rich (PR) N-terminal domain and a conserved eight-cysteine-motif (8 CM) C-terminal domain in a specific order (-C-C-CC-CXC-C-C-) [9,10]. In diverse plant species, the *HyPRP* genes play various functional roles in specific developmental stages and in responses to biotic and abiotic stresses. The strawberry *HyPRP* gene plays a putative role in anchoring polymeric polyphenols in the strawberry fruit during growth and ripening [11]. A study of three *HyPRP* genes in tobacco (*Nicotiana tabacum*) BY-2 cells indicated the involvement of the C-terminal domains of HyPRPs proteins in cell expansion [12]. A *Medicago falcata HyPRP* is induced by cold and dehydration, and expression of the *MfHyPRP* gene enhances the abiotic stress resistances in tobacco [13]. The heterologous expression of the *Cajanus cajan HyPRP* in rice increases its tolerance to abiotic and biotic stresses [14]. The *HyPRP* gene *JsPRP1* from walnut confers biotic and abiotic stresses in transgenic tobacco plants [15]. The *CaHyPRP1* gene performs distinct dual roles as a negative regulator of basal defense and a positive regulator of cell death in *Capsicum annuum* against *Xanthomonas campestris* [16]. The *HyPRP* gene *EARLI1* (Early Arabidopsis Aluminum Induced 1) improves the freezing survival of yeast cells and has an auxiliary role for germination and early seedling development in *Arabidopsis* under low temperature and salt stress conditions [17]. The *GhHyPRP4* gene is involved in the cold stress response of *Gossypium hirsutum* [18]. Overexpression of *CcHyPRP* from *Cajanus cajan* in yeast and *Arabidopsis* confers increased tolerance to drought, high salinity, and heat stresses [19]. The *GbHyPRP1* gene in *Gossypium barbadense* negatively regulates cotton resistance to *Verticillium dahliae* by the thickening the cell wall and accumulating ROS [20]. In tomato, *THyPRP* acts as a master regulator of flower abscission competence in response to ethylene signals [21], and *HyPRP1* is a negative regulator of salt and oxidative stresses [22–24].

Tomato (*Solanum lycopersicum*) is one of the world's most important vegetable crops and major fresh and processed fruit worldwide. In our previous study [22], the *HyPRP1* gene was suppressed by multiple stresses and was found to play a negative role in salt stress tolerance in tomato; moreover, $SO_2$ detoxification-related enzymes including SO, Fds, and Msr A can interact with HyPRP1. However, the role of *HyPRP1* in $SO_2$ toxicity tolerance should be further explored. In the present work, the gene features of *HyPRP1* from drought-sensitive species *S. lycopersicum* (*SlHyPRP1*) and drought-resistant species *Solanum pennellii* (*SpHyPRP1*) were compared and analyzed, and we provide evidence that it can negatively regulate the resistance to $SO_2$ toxicity by scavenging hydrogen peroxide ($H_2O_2$).

## 2. Results

### 2.1. Gene Features of SlHyPRP and SpHyPRP

We obtained a *HyPRP1* gene that was suppressed by multiple stresses in tomato from our previous experiments [22,25]. Phylogenetic analysis demonstrated that HyPRP1 from tomato shows the closest relationship to those form *Glycine max* and *Vitis vinifera* and has a more distant relationship to genes from *Arabidopsis thaliana* and *Thellungiella halophila* (Figure 1). Bioinformatics analysis showed that both SlHyPRP1 and SpHyPRP1 are comprised of 262 amino acids, which share 96.45% similarity (Figure 2A). The amino acid sequences of HyPRP1 were subjected to a BLASTP search in the SGN annotation group release proteins to identify more HyPRP in tomato. Interestingly, only one *HyPRP* gene was found in the tomato genome (data not shown).

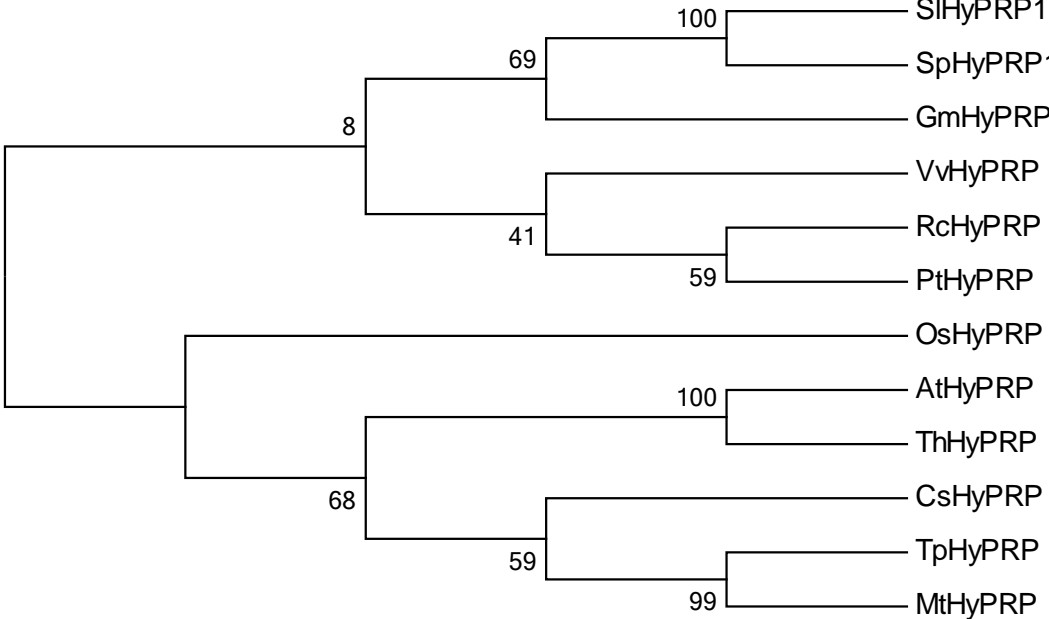

**Figure 1.** Phylogenetic analysis of HyPRP1 and homologous gene sequences from ten representative species. Species and GenBank accession numbers are shown as previously described [22]. The phylogenetic tree obtained by the neighbor-joining method and 1000 bootstrap replicates. The scale indicates branch length, and bootstrap values represent the divergence of each branch.

The DNA methylation analysis suggested that promoter and the coding sequences of *SlHyPRP1* are highly methylated. In the promoter regions, the methylation level is higher in the flower and fruit than in the leaf. However, the methylation level is significantly higher in the leaf in the coding sequences (Figure 2B and Supplementary Table S1).

In order to further study the promoter *cis*-elements, the promoter regions of *SlHyPRP1* and *SpHyPRP1* were retrieved (Supplementary file S1) and submitted to the PlantCARE database for cis-element identification. Totals of 31 and 26 *cis*-elements were identified in the *SlHyPRP1* and *SpHyPRP1* promoter regions, respectively (Table 1). The nine different *cis*-acting elements can be divided into four groups. The ABRE3a and ABRE4 *cis*-elements involved in abscisic acid (ABA) responsiveness (Group 1) appear in *SlHyPRP1* but not in the *SpHyPRP1* promoter region. The DRE was involved in dehydration, low-temperature, and salt stresses (Group 2) as well as the AT1-motif; the ATCT-motif, and chs-CMA1a cis-elements that are parts of a light responsive element (Group 3) appear in *SlHyPRP1* but not in the *SpHyPRP1* promoter region. However, the *cis*-elements of the light responsive element GA-motif (Group 3) and the auxin-responsive TGA-element (Group 4) appear in *SpHyPRP1* but not in the *SlHyPRP1* promoter region (Table 1).

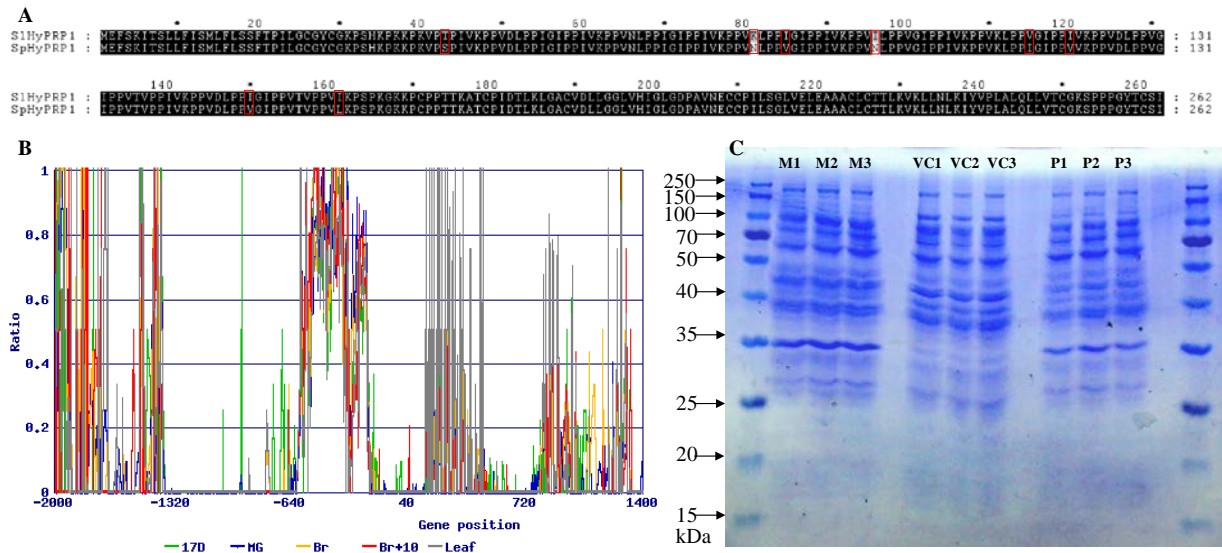

**Figure 2.** Methylation analysis and heterologous expression of *HyPRP1*. (**A**) Peptide alignment of HyPRP1 proteins of SlHyPRP1 and SpHyPRP1. (**B**) Analysis of the DNA methylation of *SlHyPRP1* at leaf and different fruit stages. Immature (17 D), mature green (MG), breaker (Br), and red ripe (Br + 10), and leaf. Ratio = 5mC/(5mC + C). (**C**) *SlHyPRP1* and *SpHyPRP1* were expressed in *Escherichia coli* and the isopropyl-1-thio-β-galactopyranoside (IPTG) induced protein product was electrophoretic separation in a 10% polyacrylamide gel. The *E. coli* cells harboring pET-*SpHyPRP1* (*HyPRP1* from *Solanum pennellii*: named P), pET-*SlHyPRP1* (*HyPRP1* from *Solanum lycopersicum* M82: named M), and control pET-E1 (vector control: named VC). Three different clones of each transformed *E. coli* were analyzed.

**Table 1.** The differences for the *cis*-elements in the promoter sequence of *HyPRP1* between *Solanum pennellii* and M82.

| M82-*HyPRP1* | *S. pennellii*-*HyPRP1* | Functions of *cis*-elements | Groups |
|---|---|---|---|
| AAGAA-motif | AAGAA-motif | | |
| ABRE | ABRE | | |
| ABRE3a | | involved in the abscisic acid responsiveness | 1 |
| ABRE4 | | involved in the abscisic acid responsiveness | 1 |
| ARE | ARE | | |
| AT-rich sequence | AT-rich sequence | | |
| AT1-motif | | part of a light responsive module | 3 |
| ATCT-motif | | involved in light responsiveness | 3 |
| AT~TATA-box | AT~TATA-box | | |
| Box 4 | Box 4 | | |
| CAAT-box | CAAT-box | | |
| CGTCA-motif | CGTCA-motif | | |
| DRE | | involved in dehydration, low-temp, salt stresses | 2 |
| ERE | ERE | | |
| G-box | G-box | | |
| | GA-motif | part of a light responsive element | 3 |
| GT1-motif | GT1-motif | | |
| MYB | MYB | | |
| MYB-like sequence | MYB-like sequence | | |
| MYC | MYC | | |
| P-box | O2-site | | |
| STRE | STRE | | |
| TATA | TATA | | |
| TATA-box | TATA-box | | |
| TATC-box | TATC-box | | |

**Table 1.** *Cont.*

| M82-*HyPRP1* | *S. pennellii*-*HyPRP1* | Functions of *cis*-elements | Groups |
|---|---|---|---|
|  | TGA-element | auxin-responsive element | 4 |
| TGACG-motif | TGACG-motif |  |  |
| Unnamed__4 | Unnamed__4 |  |  |
| Unnamed__6 | Unnamed__6 |  |  |
| WUN-motif | WUN-motif |  |  |
| as-1 | as-1 |  |  |
| chs-CMA1a |  | part of a light responsive element | 3 |

Yellow color means the *cis*-elements appear in the *HyPRP1* promoters of M82, but not in *S. pennellii*; Blue color means the *cis*-elements appear in the *HyPRP1* promoters of *S. pennellii*, but not in M82.

### 2.2. Expression of Tomato HyPRP1 Gene in E. coli Reduces Abiotic Stress Tolerance

Based on the analysis using plant cDNAs expressed in *E. coli*, the transformants enhance the host abiotic stress [26–29]. Therefore, we evaluated the tolerance of the host *E. coli* cells harboring these proteins to salinity and osmotic stresses. Under the regulation of the $T_7$ promoter, *E. coli* cells containing *SlHyPRP1* and *SpHyPRP1* expressed a polypeptide of approximately 30 kDa, which was absent in the cells transformed with the empty vector (Figure 2C). The *E. coli* cells harboring pET-*SpHyPRP1* (*HyPRP1* from *S. pennellii*: P), pET-*SlHyPRP1* (*HyPRP1* from *S. lycopersicum* M82: M), and control pET-E1 (vector control: VC) were subjected to stresses treated by NaCl and mannitol.

Under control (stress-free) media, the growth patterns of pET-*SpHyPRP1* and pET-*SlHyPRP1 E. coli* were similar to those of empty vector *E. coli* cells harboring pET-E1 (Figure 3A). Under salinity and osmotic stresses, the *E. coli* cells that expressed *SlHyPRP1* exhibited noticeably reduced resistance. However, the cells that expressed *SpHyPRP1* only showed slightly reduced tolerance to salinity and osmotic stresses (Figure 3A–C). These results indicate that *SlHyPRP1* and *SpHyPRP1* negatively regulate salinity and osmotic stress resistance in *E. coli* and exhibit distinct effects.

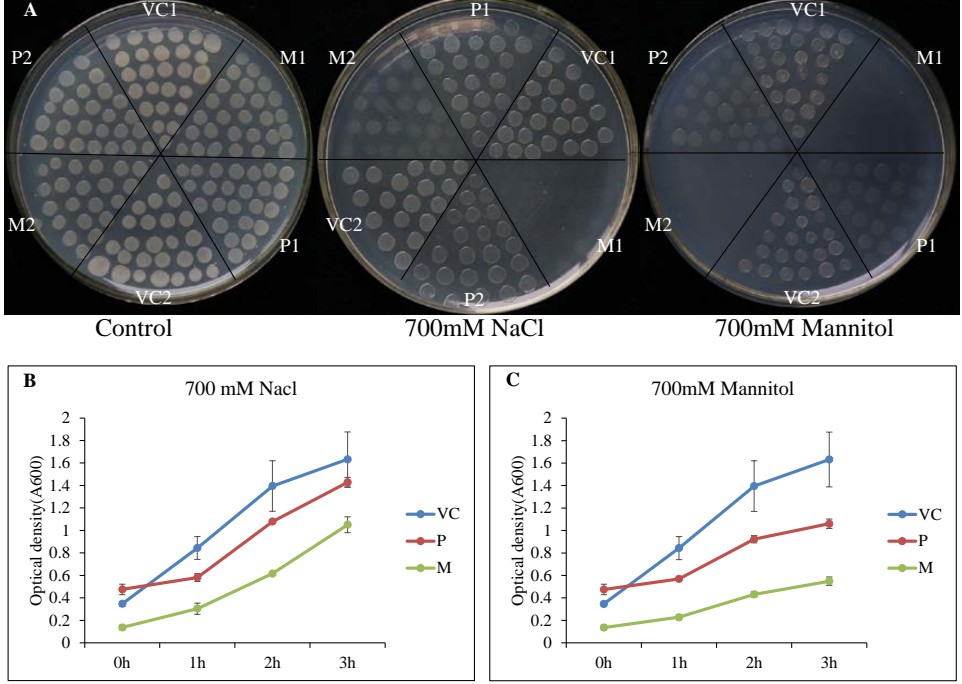

**Figure 3.** Expression of tomato *HyPRP1* in *Escherichia coli* reduces abiotic stress tolerance. (**A**) Colony growth of *E. coli* cells harboring pET-*SpHyPRP1* (P), pET-*SlHyPRP1* (M), and pET-E1 (VC) under control (stress-free), salt (700 mM NaCl) and osmotic stress (700 mM mannitol) conditions in solid LB media. (**B**) Colony growth *E. coli* cells harboring P, M, and VC under salt in liquid LB media. (**C**) Colony growth *E. coli* cells harboring P, M, and VC under osmotic stress in liquid LB media.

## 2.3. Subcellular Localization of SpHyPRP1

The putative cell-wall protein HyPRP is a dynamically evolving protein [9]. To determine whether HyPRP1 in tomato was located in the cell wall, we fused the full-length open reading frame (ORF) of *SpHyPRP1* with the N-terminal of the GFP reporter protein driven by the cauliflower mosaic virus 35S promoter (CaMV35S) promoter. The resulting fusion protein *SpHyPRP1-EGFP* was transformed into tobacco suspension cells for sub-cellular localization. However, microscopic visualization demonstrated that the green fluorescence in the transformed cell was exhibited predominantly in the cytoplasm and plasma membrane but not in the cell wall, whereas no fluorescence was detected in non-transformed cells (Figure 4A–C). Green fluorescence was exclusively detected in the entire cell region (Figure 4D–F) in these cells when only the GFP plasmid was transformed tobacco cells [30]. Hence, SpHyPRP1 is localized to the cytoplasm and plasma membrane in BY-2 tobacco suspension cells.

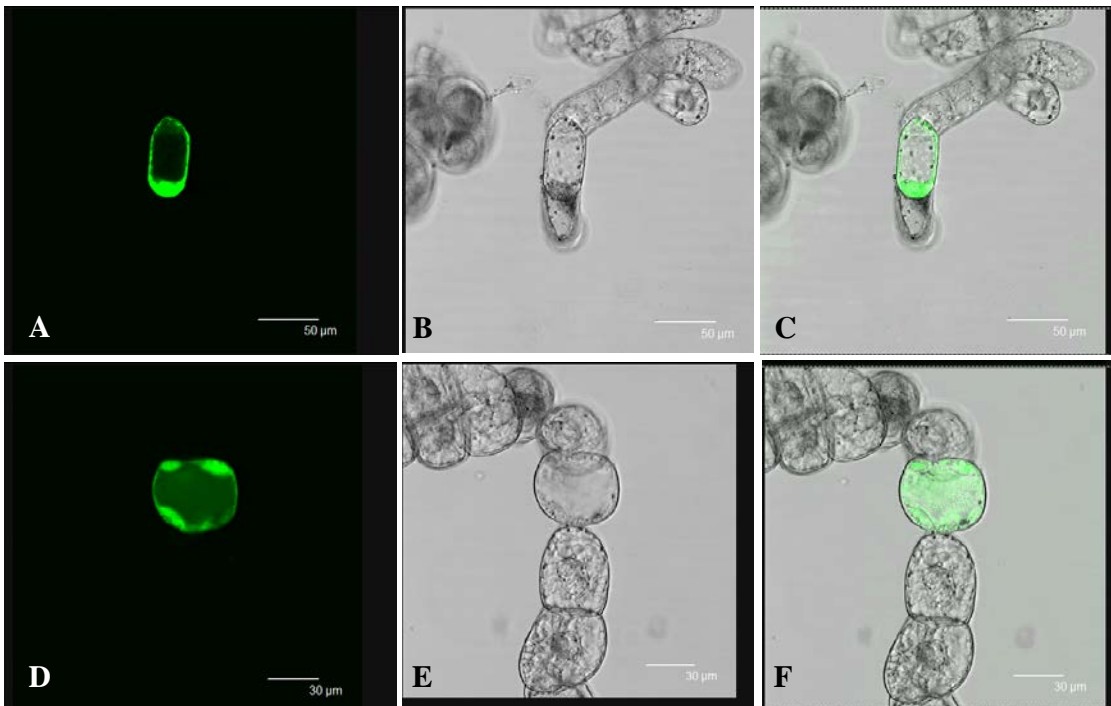

**Figure 4.** Subcellular localization of SpHyPRP1 protein in tobacco BY-2 cells. Tobacco BY-2 transiently transformed with construct containing vector plasmid *35S:: SpHyPRP1 -EGFP* (**A–C**) and *35S:: EGFP* (**D–F**). Images (**A,D**) are dark field and (**B,E**) are bright field and (**C,F**) combined.

## 2.4. Enhanced Tolerance of HyPRP1-RNAi Lines to $SO_2$

The HyPRP1 binds to the SO and Fds, which can detoxify $SO_2$; in this regard, we further investigated the potential role of *HyPRP1* to cope with sulfite or $SO_2$. Leaf discs of OE (Over-expression of *SpHyPRP1*) lines treated with 7 mM $Na_2SO_3$ showed significantly higher chlorosis and damage than wild-type and RNAi lines (Figure 5A). After treatment, the chlorophyll levels in RNAi lines decreased by 17% and 13% compared with 31% and 45% in the wild type and OE lines, respectively (Figure 5B). Hence, *HyPRP1* increases sensitivity of tomato to sulfite.

In heavily polluted geographical regions, $SO_2$ can reach a level of 2 ppm [5]; therefore, 2 ppm $SO_2$ was used for treatment. When the *HyPRP1*-RNAi lines together with their WT plants were tested with 2 ppm $SO_2$ for 2 h, the accumulation of $H_2O_2$ in situ was less intense in the *HyPRP1*-RNAi lines comparing to the WT and OE plants (Figure 6A,B). After treatment, significantly higher chlorophyll contents were retained in the *HyPRP1*-RNAi plants than in the WT and OE plants (Figure 6C), and serious damage was observed in the leaves of WT plants after 4 d (Figure 7A). The percentage of leaf damage area of RNAi

knockdown lines was only 15%, but the corresponding percentage for WT plants was 45% (Figure 7B). Similar to other abiotic stress, the toxicity of $SO_2$ enhances the ROS production by imposing oxidative stresses, and MDA is a biomarker for oxidative stress [6,31,32], therefore the content of MDA was determined. Under the control growth condition, the MDA concentration was similar to that in transgenic and WT plants. After $SO_2$ treatment, the MDA content increased in the $SO_2$-treated tomato plants, and higher levels of $H_2O_2$ accumulated in the WT and OE plants than in the Ri plants (Figure 7C).

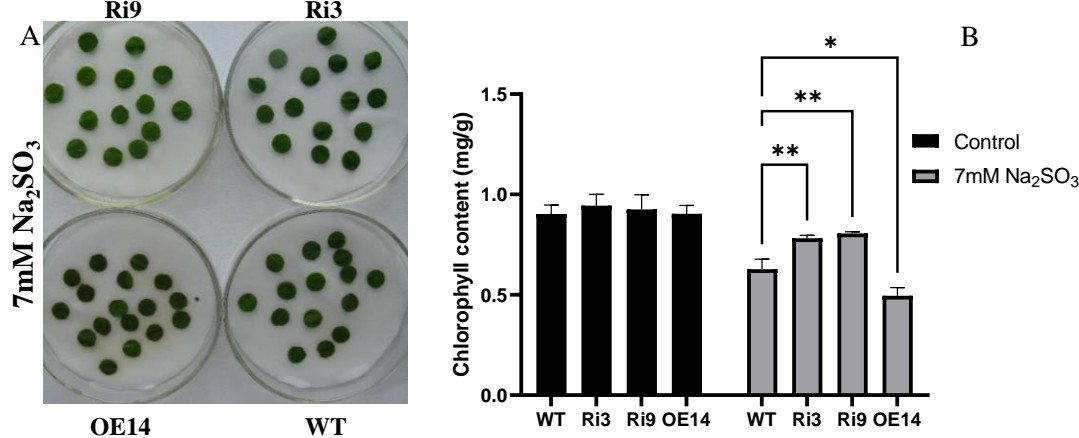

**Figure 5.** Responses of wild-type and *HyPRP1* transgenic lines treated to $Na_2SO_3$. (**A**) Representative photographs of leaf discs from wild type (WT), *HyPRP1* RNAi (Ri) and overexpression plants (OE) were treated with 7 mM $Na_2SO_3$ for 24 h. (**B**) Chlorophyll content of the no-stress (Control) and $Na_2SO_3$-treated leaf discs. Means SE (*n* = 3). The difference between OE or Ri compared to WT indicated * *p* < 0.05, ** *p* < 0.01.

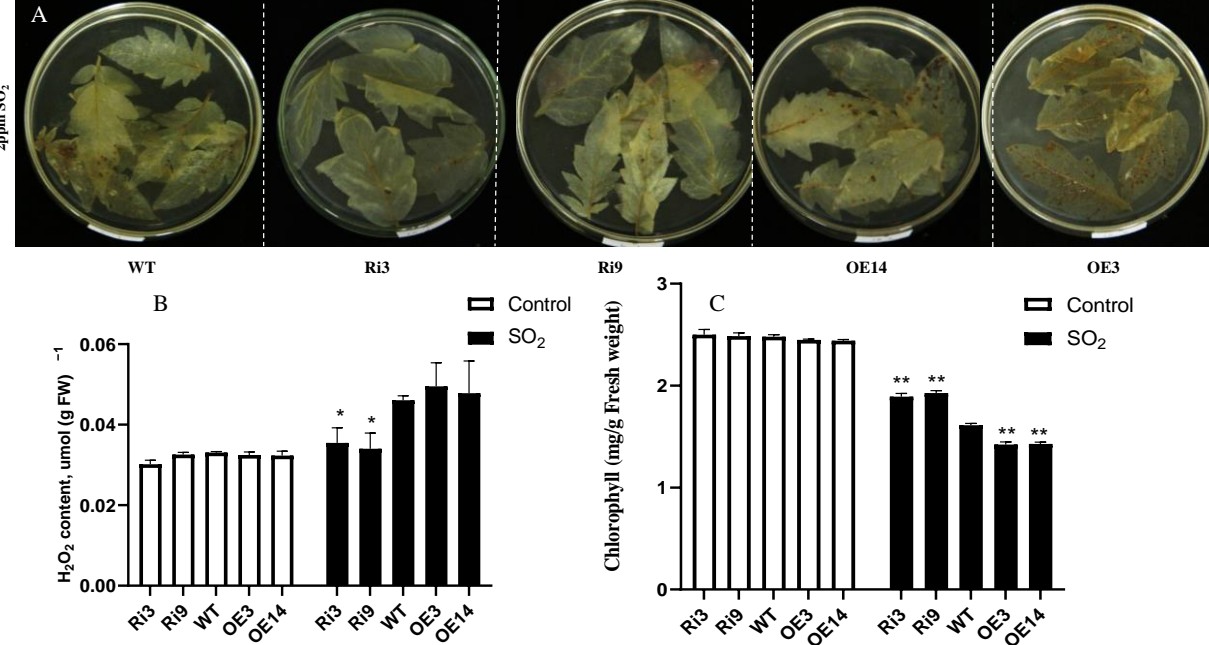

**Figure 6.** Accumulation of $H_2O_2$ and chlorophyll in *HyPRP1* RNA interference lines (Ri), overexpression lines (OE) and wild-type (WT) plants exposed to $SO_2$ toxicity. (**A**) Accumulation of $H_2O_2$ after $SO_2$ exposure at 2 ppm for 2 h in leaves staining with DAB. (**B**) $H_2O_2$ concentration in 2 ppm $SO_2$-treated for 2 h and untreated (Control) leaves from Ri, OE, and WT plants. (**C**) Chlorophyll content of the no-stress (Control) and 2 ppm $SO_2$-treated for 2 h leaves from Ri, OE, and WT plants. Values are means ± SE (*n* = 3). The difference between OE or Ri compared to WT indicated * *p* < 0.05, ** *p* < 0.01.

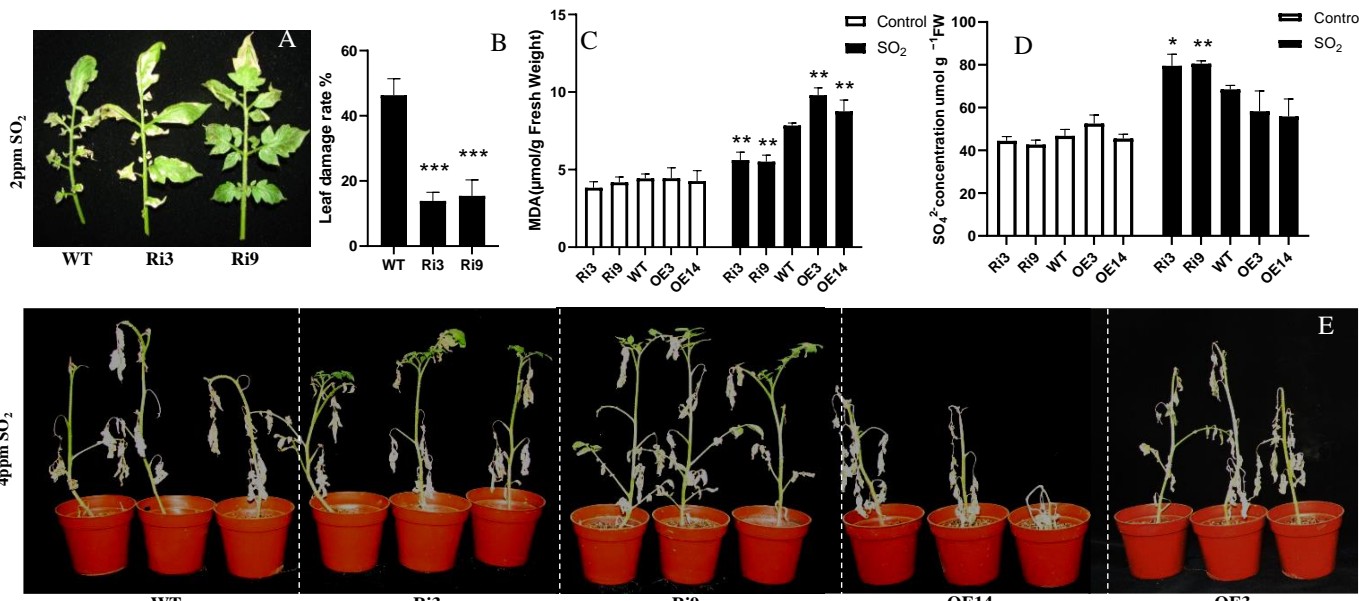

**Figure 7.** Responses of the *HyPRP1* RNA interference lines (Ri), overexpression lines (OE) and wild-type (WT) plants to $SO_2$ toxicity at different concentrations. (**A**) Leaves of treated WT and Ri plants which recovered 4 d after $SO_2$ exposure at 2 ppm for 2 h. (**B**) Effect of $SO_2$ on Ri and WT plants which recovered 4 d after exposure as (**A**) measured by leaf damage area (%) (The percentage of damaged leaf area in whole leaf area, sampled at the fourth and fifth leaves from the top of the plant). Values are means $\pm$ SE ($n$ = 12). (**C**) MDA concentration in 2 ppm $SO_2$-treated for 2 h and untreated (Control) leaves from Ri, overexpressed, and WT plants. Values are means $\pm$ SE ($n$ = 3). (**D**) Sulfate concentration in 2 ppm $SO_2$-treated for 2 h and untreated (Control) leaves from Ri, overexpressed, and WT plants. Values are means $\pm$ SE ($n$ = 3). (**E**) The growth of *HyPRP* RNAi (Ri) and wild-type (WT) plants which recovered 7 d post-exposure to 4 ppm $SO_2$ for 4 h. The difference between OE or Ri compared to WT indicated * $p < 0.05$, ** $p < 0.01$, *** $p < 0.001$.

In contrast to ROS, $SO_2$ can be rapidly transformed into sulfite in plant leaves, which is the main component of acid rain and can cause direct oxidative stress [33]. Sulfite oxidase (SO; EC 1.8.3.1) can oxidize sulfite to non-toxic sulfate, and the sulfites are rapidly oxidized to sulfate during extraction [34]. Thus, the sulfate contents in the transgenic lines were determined. Before $SO_2$ treatment, the sulfate content in the transgenic and WT plants had no significant difference but was significantly lower than that after $SO_2$ treatment. After $SO_2$ treatment, the sulfate content was higher in the *HyPRP1*-RNAi lines but lower in the WT and OE lines (Figure 7D). These results indicate that the *HyPR1P*-RNAi plants can catalyze the conversion of sulfite to non-toxic sulfate when plants are subjected to $SO_2$ pollution.

When the plants were exposed to 4 ppm $SO_2$ for 4 h, more than 80% *HyPRP1*-RNAi lines recovered within 7 d, but all *HyPRP1*-overexpressing and WT plants failed to recover (Figure 7E). This finding demonstrates that the detoxifying capacity of $SO_2$ was significantly enhanced by scavenging $H_2O_2$ in the *HyPRP1*-RNAi plants.

### 2.5. OE of the SpHyPRP1 in Tobacco Reduces Salt Stress Resistance

To determine if *SpHyPRP1* is the basis of abiotic resistance and acts in a similar manner in other species, we transformed the *35S:: SpHyPRP1* into tobacco for functional analysis. The kanamycin-tolerant tobacco plants were PCR confirmed using 35S and *SpHyPRP1* gene reverse primers. After RT-PCR analysis, the significantly over-expressed *SpHyPRP1* lines OE2, OE5, and OE51 were selected for further functional analysis. The tolerance exhibited by the $T_1$ transgenic lines to salinity stress was analyzed by subjecting them to NaCl stress. Under 200 mM NaCl stress in the filter paper, the seedlings of WT grew well, but OE plants did not survive (Figure 8A). For further functional analysis, the uniform seedlings were

treated with different concentrations of NaCl. Under unstressed conditions, the WT and OE lines showed similar growth pattern. The OE lines revealed noticeable decreases in plant growth and chlorophyll content compared with the wild-type plants (Figure 8B–D) under 150 and 200 mM NaCl. The transgenic OE lines could not survive under 300 mM NaCl stress compared with the wild-type plants and showed observable decreases in plant growth and chlorophyll content (Figure 8).

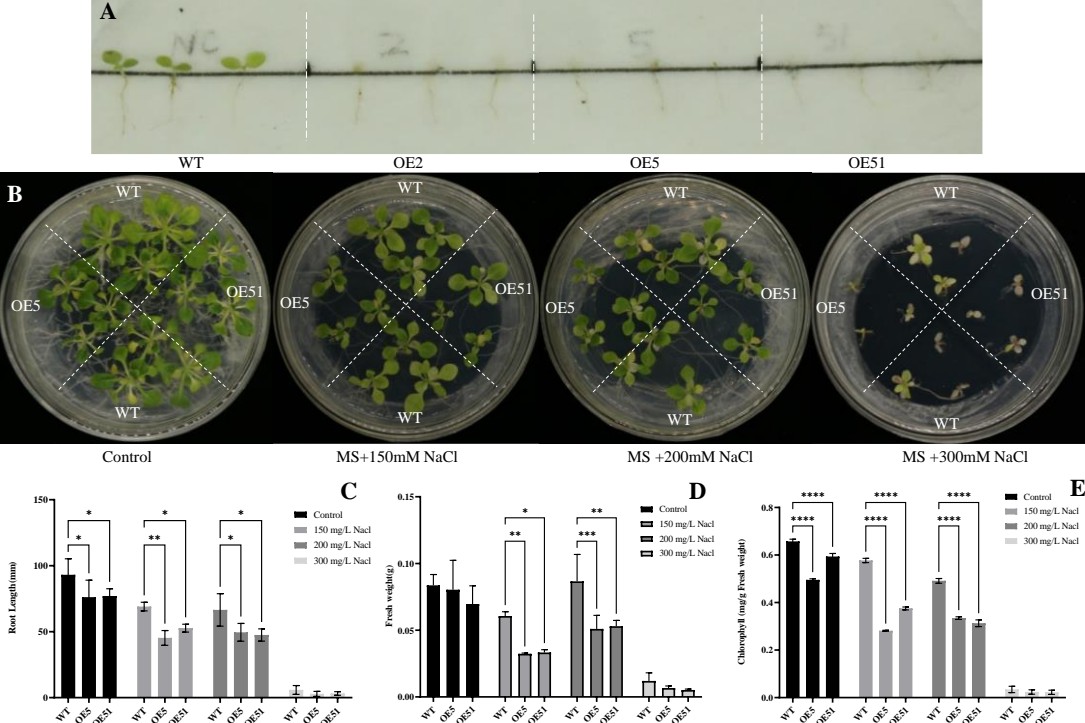

**Figure 8.** Determination of salt tolerance of *HyPRP1*-overexpressing (OE) tobacco plants. (**A**) Salt resistance testing of *HyPRP1* OE tobacco seedlings treated by 200 mM NaCl. *HyPRP1* OE T$_1$ seedlings after kanamycin selection shifted to the filter with MS by adding 200 mM NaCl for treatment. (**B**) Phenotype of WT and *HyPRP1* OE tobacco seedlings treated by 150, 200, and 300 mM NaCl. Uniformed transgenic T$_2$ seedlings were selected and shifted to MS media by adding 150, 200, and 300 mM NaCl for treatment respectively. Changes in root length (**C**), fresh weight (**D**) and chlorophyll content (**E**) in response to salt stress in WT and HyPRP1 OE tobacco seedlings. Values are means ± SE (*n* = 3). The difference between OE or Ri compared to WT indicated * $p < 0.05$, ** $p < 0.01$, *** $p < 0.001$, *** $p < 0.0001$.

## 3. Discussion

Plants encounter different biotic or abiotic stresses during their life cycle in a fixed place. Therefore, plants have acquired the capability of adapting to adverse circumstances owing to changed gene expression profiles and rapid signaling in response to stress [35]. Therefore, understanding environment responses is one of the most important topics in plant science. At present, many abiotic stress-related genes have been functionally identified, which can improve the resistance of plants against abiotic stresses [36]. In the present study, the functional role of *HyPRP1* in imparting multiple abiotic stress tolerance was determined by genetic modification in tomato, *E. coli*, and tobacco systems.

Sulfur dioxide, a major atmospheric contaminant when dissolved in water, can transform into sulfite, which can cause direct oxidative stress [33]. Sulfite also induces sulfitolysis and inactivates proteins, such as thioredoxins [37]. The HyPRP1 physically interacts with several abiotic response genes (SO, Fds, and Msr A) [22], suggesting that HyPRP1 acts along a signaling pathway and not as a final component. The SO catalyzes the conversion of sulfites to non-toxic sulfate to cope with sulfite overflow [5]. The Fds is an electron donor

for sulfite reductase that catalyzes the reduction of sulfite into sulfide [38]. The MsrA can reduce methionine sulfoxide residues in proteins formed by the oxidation of methionine by ROS [39]. Silencing of the *HyPRP1* gene results in accumulating more transcripts of *Msr A* and *Fds* than wild-type plants under $SO_2$ stress [22]. In the present study, the suppression of *HyPRP1* significantly induces the tolerance to $SO_2$ toxicity by scavenging $H_2O_2$ and sulfite and reduced the oxidative damage and chlorophyll destruction (Figures 6 and 7). Based on these observations, we suggest that *HyPRP1* negatively modulates *Msr A*, *SO*, and *Fds* to confer oxidative and $SO_2$ toxicity by compromising the scavenging of hydrogen peroxide and sulfite.

The difference in the stress tolerance phenotypes of *SpHyPRP1* and *SlHyPRP1* expressed in *E. coli* indicates that *HyPRP1* from the two tomato species shows different effects on abiotic resistance (Figure 3). In the promoter regions, the main different *cis*-element of *HyPRP1* between cultivated drought-sensitive and wild drought-resistant tomato is the ABA-responsive element (Table 1). The phytohormone ABA plays an essential role in adaptive stress responses via the ABA-responsive *cis*-element in the promoters of many abiotic stress-inducible genes [40]. Hence, *SpHyPRP1* and *SlHyPRP1* form different evolutionary materials and exhibit different functions via ABA response by the *cis*-element. In the promoter regions, the methylation level of *HyPRP1* is significantly higher in the flower and fruit than in the leaf (Figure 2B and Supplementary Table S1). This finding can explain why *HyPRP1* is significantly highly expressed in tomato leaves [22].

The HyPRPs are considered to be putative cell wall proteins with distinct amino acid characteristics [12]. In this study, The tomato HyPRP1 contain a proline-rich signal peptide and a conserved 8CM domain [22]. Although the CcHyPRP protein in *Cajanus cajan* has been reported to be a cell-wall protein [14], we found that SpHyPRP1 is a cytoplasm and plasma membrane protein when transiently expressed in tobacco cells (Figure 4). This is consistent with the observation that SlHyPRP1 and DEA1 localize and interact in the cytoplasm and plasma membrane in vivo [41]. These indicate that HyPRP1 in various plant species are divergent cell-specific features.

Plant *HyPRPs* genes have reported to be responsive to abiotic factors. According to our recent research, the expression of tomato *HyPRP1* is inhibited by drought, high salinity, cold, heat, and oxidative stresses [22]. In cotton, a gene encoding putative HyPRP, named *GhHyPRP4*, was significantly upregulated in the leaves of cotton seedlings under cold stress. When the *GhHyPRP4* promoter drives the GUS (b-glucuronidase) gene in transgenic *Arabidopsis thaliana*, the gene was specifically expressed in the leaves and cotyledons and remarkably induced by cold stress [18]. In rice, 45 OsHyPRP genes were identified, and their transcriptional responses to biotic and abiotic stresses and hormone treatment were analyzed [42]. The recent functional analysis of HyPRPs suggests their diverse roles in biotic and abiotic stress tolerance [13–18,20–22,43–45]. Moreover, the heterologous expression of *HyPRP* in *E. coli* and tobacco reduce their abiotic resistance (Figures 3 and 8), and HyPRP1 is widely present in plants (Figure 1); it implies that the *HyPRP1* is fundamental to abiotic resistance. All the above studies highlight the importance of *HyPRPs* in biotic and abiotic responses; therefore, identifying and characterizing *HyPRPs* in plants are of great significance.

The present findings together with our previous results [22] indicate that tomato *HyPRP1* gene negatively regulates the stress response against abiotic stresses in tomato, tobacco, and *E. coli*. The CRISPR/Cas9 system is increasingly used in plants to create gene mutants that can be used to improve plants [46,47]. Silencing of the negative regulator *HyPRP1* can mitigate $SO_2$ toxicity and enhance oxidative tolerance, which is a promising candidate for CRISPR/associated nuclease Cas9 knockout to create transgene-free, mutant abiotic resistance tomato [23,24] or other species progeny.

## 4. Materials and Methods

### 4.1. Phylogenetic Analysis of HyPRP1, Prediction of cis-element, and Methylation Analysis of HyPRP1

A phylogenetic analysis of SpHyPRP1 (SGN No. Sopen12g004640.1) and SlHyPRP1 (Solyc12g009650.1.1) with their homologous gene was constructed using MEGA (version 7) software [48]. The full length of the *SpHyPRP1* and *SlHyPRP1* sequences were used as BLASTN search queries against the *Solanum pennellii* WGS Chromosome data and tomato whole genome scaffolds (version 2.40) at the SGN website (https://solgenomics.net/, accessed on 1 January 2022), respectively. The 1.5 kb promoter sequences upstream of *SpHyPRP1* and *SlHyPRP1* were retrieved and submitted to the PlantCARE database (http://bioinformatics.psb.ugent.be/webtools/plantcare/html/, accessed on 1 January 2022) for *cis*-element prediction. Protein properties were analyzed by SMS2 software (http://www.detaibio.com/sms2/protein_mw.html, accessed on 1 January 2022). The cytosine methylation ratio: $5mC/(5mC + C)$ of the *SlHyPRP1* promoter sequence and coding sequence in various tomato tissues was analyzed using the epigenetic database of tomatoes (http://ted.bti.cornell.edu/epigenome/, accessed on 1 January 2022).

### 4.2. HyPRP1 Expression in E. coli and Abiotic Stress Treatment

The constructs of pET-*SpHyPRP1*, pET-*SlHyPRP1*, and the empty vector pEASY-E1 in *E. coli* BL21 (DE3) cells [22] were used to measure the growth rate under salt and simulated drought stress conditions. Gene expression in *E. coli* cells harboring pET-*SpHyPRP1*, pET-*SlHyPRP1*, and control pET-E1 was induced by isopropyl-1-thio-β-galactopyranoside (IPTG) as previously described [22]. The induced protein product was electrophoretically separated on 10% polyacrylamide gel. The induced cells were treated with 700 mM NaCl or mannitol in the liquid LB media. After 1 h, 2 h, and 3 h, the $OD_{600}$ was measured. In brief, 2 μL of the cells were dotted in the solid LB media challenged with 700 mM NaCl or 700 mM mannitol for 4 h.

### 4.3. Subcellular Localization of SpHyPRP1

For subcellular localization analysis, the coding region (without stop codon) of *SpHyPRP1* was obtained by PCR amplification using tomato cDNA as a template with the following primers (forward primer 5′-<u>GTCGAC</u>ATGGAGTTCTCTAAGATAACTTC-3′, reverse primes 5′-<u>GGTACC</u>GATGGAACAAGTGTAGCCAGG-3′, the underline is restriction enzyme cutting site. After sequencing confirmation, the PCR product was digested from the T vector then inserted into the CaMV35S with EGFP (enhanced green fluorescent protein) fusion construction. The resulting *35S:: SpHyPRP1 -EGFP* fusion construct was bombarded into BY-2 tobacco suspension cells by using Biolistic PDS-1000 (Bio-Rad, Hercules, CA, USA). All samples were observed under a Confocal Laser Microscopy (Zeiss, LSM780, Oberkochen, Germany) after 24 h of infiltration.

### 4.4. Leaf Disc and SO$_2$ Treatment of HyPRP1 Transgenic Plants

For leaf disc treatment, discs (9 mm diameter) were cut from 5-week-old wild-type and transgenic tomato youngest fully-expanded leaves and immediately placed in 90 mm-diameter plates with 50% MS salt solution [5]. The leaf discs were placed on a filter paper moistened with or without 2 mL of 7 mM $Na_2SO_3$($SO_2$ donor) for 24 h under constant lighting (100 μmol m$^{-2}$ s$^{-1}$) and then analyzed for chlorophyll content and photographed. Chlorophyll content was measured by Lichtenthaler method [49].

For tomato (*S. lycopersicum*), *HyPRP1*-RNAi suppression (Ri) and overexpression (OE) lines in the cv. M82 background were developed previously [22]. The progeny from independent transformation events was used in further experiments. The $SO_2$ exposure and sulfate content was carried out as described previously [22]. The percentage of leaf damage area was calculated as the ratio of damage leaf area (measuring by transparent lattice squares) divided by the whole leaf area of the treated plants and multiplied by 100. The $H_2O_2$ level was determined according to the method described previously [50]. The $H_2O_2$

in the leaves in situ was examined by histochemical staining with 3, 3′-diaminobenzidine (DAB) [22]. Malondialdehyde (MDA) was assayed for indirect evaluation of lipid peroxidation using thiobarbituric acid as previously described [51].

*4.5. Salt Stress Resistance Analysis of Heterologous Expression HyPRP1 in Tobacco*

For further investigate the features of *HyPRP1*, the full length of *HyPRP1* ORF was cloned into the binary vector pBI121 through the CaMV35S to yield the overexpressing construct. The construct was transformed to tobacco (*Nicotiana nudicaulis*) by *Agrobacterium tumefaciens* (strain LB4404)-mediated transformation. The T$_1$ transgenic tobacco plants were PCR confirmed, and the expression of *HyPRP1* in the over-expressed transgenic T$_1$ plants was investigated by real-time PCR as described previously [22].

Positive transgenic tobacco seedlings were germinated in a Petri dish with filter paper for 4 days to evaluate the salt tolerance. The uniform germinated seeds were sub-cultured on filter paper containing 200 mM NaCl for salt treatment. The transgenic and wild type seedlings were treated for 5 days. The salt resistance investigation in MS medium test and chlorophyll content assay for T$_2$ tobacco seedlings was tested as previously described [30].

*4.6. Statistical Analysis*

The data were analyzed by Tukey's multiple comparison test in the ANOVA program of GraphPad Prism 9.0, $p < 0.05$ (*), $p < 0.01$ (**), $p < 0.001$ (***) indicates significantly different. All the data of abiotic stress treatment are from one of two or three different experiments that yielded essentially identical results.

**5. Conclusions**

We performed functional analysis of the tomato *HyPRP1* gene in response to abiotic stress by expressing it in tomato, tobacco, and *E. coli*. The HyPRP1 protein plays a negative role in regulation of the resistance to SO$_2$ toxicity modulated by *SO*, *Fds*, and *Msr A* and resistance to salinity and osmotic stresses. The difference in the stress-tolerance phenotypes of *SpHyPRP1* and *SlHyPRP1* expressed in *E. coli* indicates that *HyPRP1* from the two species shows different effects on abiotic resistance. Hence, *HyPRP1* will be a desired candidate for gene editing mutation for plant abiotic resistance improvement.

**Supplementary Materials:** The following supporting information can be downloaded at https://www.mdpi.com/article/10.3390/horticulturae8121118/s1: Supplementary file S1 Promoter sequences of SlHyPRP1 and SpHyPRP1 as showed in red; Table S1 The methylation of *HyPRP1* in tomato different tissues and stages.

**Author Contributions:** Investigation, data curation, and formal analysis, X.C.; investigation and visualization, Y.L. and X.H.; conceptualization and resources, L.W.; investigation and visualization Q.P.; software and validation, Y.D.; funding acquisition, writing—review and editing, J.L. All authors have read and agreed to the published version of the manuscript.

**Funding:** This work was supported by grants from the National Natural Science Foundation of China (No. 31872123), the Natural Science Foundation of Chongqing, China (No. cstc2019jcyj-msxmX0333), and Fundamental Research Funds for the Central Universities (No. XDJK2020B060).

**Data Availability Statement:** Data are contained within the article.

**Conflicts of Interest:** The authors declare no conflict of interest.

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
