# Peer review of "HyPRP1, A Tomato Multipotent Regulator, Negatively Regulates Tomato Resistance to Sulfur Dioxide Toxicity and Can Also Reduce Abiotic Stress Tolerance of Escherichia coli and Tobacco"

_horticulturae, doi:10.3390/horticulturae8121118_

Round 1

Reviewer 1 Report

Introduction

Line 48. This sentence is strangely written. I think what the authors want to say is:

Sulfite can be converted by oxidation to sulfate by the molybdene….  and by reduction to sulfide  by sulfite reductase through a process…

Similarly, line 81: I am not sure HyPRP1 is necessary in this sentence.

Same for the sentence beginning line 87. What does ‘gene suppressed by multiple stresses’ mean? Same for line 98. Give the reference of ‘our previous study’.

Results

I do not understand the sentence line100 claiming that the closest relationship between the HyPRP1 from tomato (actually 2 tomato species) is with the one from Vitis and Cucumis, while it seems that the closest is from Glycine in figure 1.

Line 109: I guess it means: ‘Interestingly, only one HyPRP gene…’

The five groups mentioned line 122 should be better identified in table 1.

Line 156: the sentence is not clear enough.

Line 204: the sentence is imprecise (‘copy with sulfate’).

Line 207: ‘the’ is not necessary. Same in line 216.

Legend to figure 5. Title: ‘to toxic level of NASO3’. A) ’…OE treated with 7mM…’

Legend to figure 6. Title: ‘…in HyPRP1 RNAi…’

Legend to figure 7. In 3 instances, ‘recovered’ should be preceded by ‘which’.

Line 280. HyPRP1 cDNA ?

Discussion

Regarding the statement around line 349 about the subcellular location of spHyPRP, since this result is solely derived from transient expression, the author should be more careful (since they do not have confirmation from stable transgenic lines).

Suggestion to improve the text:

Line 326: ‘silencing of HyPRP1 results in accumulation of more transcrits’

Line 343: ‘This finding can explain that…’

Line 349. ‘… has been reported to be a cell wall….’. ‘…we found that SpHYPRP1 is a….’

Line 352. ‘HyPRP1 in various plant species are…’

Line 358.’When the GhPRP4 promoter… thaliana, the…’

Line 365. ‘HyPRP1 is widely…’

Line 366. ‘ present in plants…’; ‘HyPRP1 is at the basis…’

Reviewer 2 Report

The correspondence author of this manuscript previously found that tomato HyPRP1 is a negative regulator of salt and oxidative stresses and is probably involved in sulfite metabolism. In this study, the authors further analyzed the DNA methylation status of this gene in different plant tissues and subcellular localization of this protein. They also found heterologous expression of tomato HyPRP1 reduced abiotic stress tolerance of E. Coli and tobacco. Moreover, the authors further provided evidence HyPRP1 negatively regulated the detoxification pathway of tomato to sulfur dioxide exposure using OE and RNAi transgenic plants. The whole study is a good supplementary support of their previous publication, indicating multiple roles and ability of HyPRP1 in abiotic stress regulation.   However, putting all these independent experiments and analysis together makes this manuscript lack logic and major points, especially in the title and abstract, which really confused me a lot. English writing is good but there are some mistakes to be improved. Please find the major and minor issues listed below.

Major issues:

As I mentioned above, this manuscript presented further evidence of the multiple function of HyPRP1 in regulating salt stress, not only in tomato but even in E. Coli and tobacco. However, the authors mainly clarify the inhibitory role of HyPRP1 in SO2 detoxification and H2O2 scavenging. But the title and abstract of this manuscript looks very confusing and lacks major points. The focus and logic of this manuscript is supposed to be adjusted. By the way, I don’t understand “negatively protect plants” means.

Minor issues:

1.      The transgenic plant OE lines and RNAi lines were used in this study. The transcriptional expression of HyPRP1 for each line is supposed to be determined by qRT-PCR. Two independent lines with different expression levels are recommended.

2.      “The Phylogenetic analysis….” at Page 3 Line 100-103 demonstrated tomato HyPRP1 has the closest relationship to Vitis vinifera and Cucumis sativus and a more distant relationship to genes from Arabidopsis thaliana and Thellungiella halophila. However, the phylogenetic tree built based on NJ method doesn’t show it clearly. It looks like they have the closest relationship to GmHyPRP1.

3.      The polypeptide alignment of SlHyPRP1 and HpHyPRP1 is slightly different from that in the authors’ previous publication (Li et al., 2016, Frontiers in Plant Science). There are 8 amino acids differences in the previous paper compared with only two spots in this manuscript. Please clarify it.

4.      Expression of HyPRP1 in E. Coli is suggested to be detected via western blot.

5.      “Plant physiological activities is initial declined appears several days when exposure to SO2” At Page 2 Line 54-55. Corrected as “Plant physiological activities decline in several days after exposure to SO2””

6.      “A study of three HyPRP genes…” at Page 2 Line 67. Gene name is in italics. Please check the same issue throughout the manuscript.

7.      “The overexpression of ….” At Page 2 Line79. Delete “The”

8.      “GbHyPRP1 in Gossypium barbadense HyPRP1” at Page 2 Line81. Corrected as “Gossypium barbadense HyPRP1(GbHyPRP1)” or “GbHyPRP1 in Gossypium barbadense”

9.      “HyPRP1 gene suppressed by multiple stresses” at Page 2 Line88. Corrected as “HyPRP1 gene was suppressed by multiple stresses”.

10.   The note “Yellow (blue) color means the cis-elements appear in the HyPRP promoters of M82 (S.pennellii), but 153 not in S.pennellii (M82).” at Page 5 Line 153-154 is confusing. Please rephrase it.

11.   “potential role of HyPRP1 copy with sulfite or SO2” at Page 7 Line 203-204. Corrected as “cope”

12.   “Leaf discs of overexpression lines……” at Page 7 Line 205. Corrected as “Leaf discs of overexpression lines treated with 7mM Na2SO3 showed….”

13.   “After treatment, the chlorophyll levels of 17% and 13% were detected in RNAi lines…” at Page 7 Line207-208. Corrected as “After treatment, the chlorophyll levels in RNAi lines decreased by 17% and 13% compared with that of 31% and 45% in wild type and OE lines, respectively.”

14.   “OE line” at Page 7 Line 208. Please clarify HyPRP1 overexpressed in tomato is Sl or Sp HyPRP1.

15.   “Air pollution caused by SO2 results in acid rain…” at Page 7 Line 210 is redundant with” In contrast to ROS, SO2 is a damaging air pollutant that can be transformed into sulfite, which is the main component of acid rain and can cause direct oxidative stress” at Line 226-227. Please reedit it.

16.   “HyPRP1 negatively modulates Msr A, SO, and Fds to confer oxidative and SO2 toxicity by scavenging hydrogen peroxide and sulfite” at Page 11 Line330-331. Corrected as “by compromising the scavenging of hydrogen peroxide and sulfite.”

17.   “This finding is consistent with the observation that ….” at Page 11 Line 334-336. The different effects on abiotic stress in E. Coli conferred by SpHyPRP1 and SlHyPRP1 doesn’t result from the different cis-element in their promoter regions. Only the CDS regions were transformed into E. coli and their expression was initiated by T7 promoter. Hence, it is inappropriate to say, “This finding is consistent with….”. These two sentences lack such correlation.

18.   “Silencing of the negative regulator HyPRP1 can enhance SO2 toxicity…” at Page 12 Line 373. Corrected as” Silencing of the negative regulator HyPRP1 can mitigate SO2 toxicity and enhance oxidative tolerance…”

Round 2

Reviewer 2 Report

Minor issue:

11.    I still think the title of this manuscript seems like a bit redundant. My suggestion is " HyPRP1, a tomato multipotent regulator, negatively regulates tomato resistance to sulfur dioxide toxicity and can also reduce abiotic stress tolerance of Escherichia coli and tobacco”.

22. Page 1, Line 18. A is a lower case.

33.  Page 2 Line1, corrected as “Sulfur dioxide (SO2) is a harmful air pollutant that has….”

44.  Page 3 Line 2-3, corrected as “In our previous study [22], the HyPRP1 gene was suppressed by multiple stresses and was found to play a negative role…”

55.  Page 3 Line 4, corrected as” SO, Fds, and Msr A can interact with HyPRP1”

66.  Page 3 Line 6, corrected as “were compared and analyzed”.

77.  Page 3 Line 7, corrected as” it can negatively regulate the resistance”

88. Page 8 Line 19, corrected as “When the plants were exposed to 4 ppm SO2 for 4 h, more than 80% …”

99. Page 11 Line 13, corrected as “the capability of adapting to adverse circumstances owing to changed gene expression profile and rapid signaling in response to stress.”

110.  Page 12 Line 3, corrected as “results in accumulating…”

111.   Page 12 Line 3-4, corrected as “…significantly induced the tolerance of tomato to SO2 toxicity by scavenging H2O2 and sulfite and reduced…”

Author Response

Dear Sir/ Madam, We are very thankful for a thorough review and for the helpful comments regarding our manuscript. We have fully revised the manuscript based on your comments. Changes made to the text using the “*Track Changes*” so that they can be easily identified. 11. I still think the title of this manuscript seems like a bit redundant. My suggestion is " HyPRP1, a tomato multipotent regulator, negatively regulates tomato resistance to sulfur dioxide toxicity and can also reduce abiotic stress tolerance of Escherichia coli and tobacco”. 22. Page 1, Line 18. A is a lower case. 33. Page 2 Line1, corrected as “Sulfur dioxide (SO2) is a harmful air pollutant that has….” 44. Page 3 Line 2-3, corrected as “In our previous study [22], the HyPRP1 gene was suppressed by multiple stresses and was found to play a negative role…” 55. Page 3 Line 4, corrected as” SO, Fds, and Msr A can interact with HyPRP1” 66. Page 3 Line 6, corrected as “were compared and analyzed”. 77. Page 3 Line 7, corrected as” it can negatively regulate the resistance” 88. Page 8 Line 19, corrected as “When the plants were exposed to 4 ppm SO2 for 4 h, more than 80% …” 99. Page 11 Line 13, corrected as “the capability of adapting to adverse circumstances owing to changed gene expression profile and rapid signaling in response to stress.” 110. Page 12 Line 3, corrected as “results in accumulating…” 111. Page 12 Line 3-4, corrected as “…significantly induced the tolerance of tomato to SO2 toxicity by scavenging H2O2 and sulfite and reduced…” Response:Thanks for your thorough editing and for the comments. Revised according to the suggestions.